# Safety and Diagnostic Efficacy of Image-Guided Biopsy of Small Renal Masses

**DOI:** 10.3390/cancers16040835

**Published:** 2024-02-19

**Authors:** Muhamad Serhal, Sean Rangwani, Stephen M. Seedial, Bartley Thornburg, Ahsun Riaz, Albert A. Nemcek, Kent T. Sato, Kent T. Perry, Bonnie Choy, Robert J. Lewandowski, Andrew C. Gordon

**Affiliations:** 1Section of Interventional Radiology, Department of Radiology, Northwestern University Feinberg School of Medicine, Chicago, IL 60611, USA; muhamad.serhal@northwestern.edu (M.S.); stephen.seedial@aiarad.com (S.M.S.); bartley.thornburg@northwestern.edu (B.T.); ahsun-riaz@northwestern.edu (A.R.); aan728@northwestern.edu (A.A.N.J.); k-sato@northwestern.edu (K.T.S.); r-lewandowski@northwestern.edu (R.J.L.); 2Northwestern University Feinberg School of Medicine, Chicago Campus, Chicago, IL 60611, USA; sean.rangwani@northwestern.edu; 3Department of Urology, Northwestern University Feinberg School of Medicine, Chicago, IL 60611, USA; kperry@northwestern.edu; 4Department of Pathology, Northwestern University Feinberg School of Medicine, Chicago, IL 60611, USA; bonnie.choy2@nm.org

**Keywords:** kidney neoplasms, image-guided, biopsy, complication

## Abstract

**Simple Summary:**

This study investigates the safety and effectiveness of using image-guided biopsy to diagnose small renal masses less than 4 cm in size. Conducted retrospectively at a single center from 2015 to 2021, it focuses on patients without a previous cancer history or larger renal masses. By analyzing patient demographics, tumor size, biopsy methods, complications, and diagnoses, we aimed to determine the reliability of biopsy as a diagnostic tool. Our findings suggest that this method is safe and provides valuable diagnostic information, particularly for renal masses between 3 and 4 cm. The results have important implications for patient management, showing a significant number of benign diagnoses and influencing treatment choices, preventing unnecessary surgical intervention in a significant number of cases. This research contributes to the growing acceptance of renal mass biopsy in clinical practice, offering insights into its benefits for patient care.

**Abstract:**

Introduction: Image-guided renal mass biopsy is gaining increased diagnostic acceptance, but there are limited data concerning the safety and diagnostic yield of biopsy for small renal masses (≤4 cm). This study evaluated the safety, diagnostic yield, and management after image-guided percutaneous biopsy for small renal masses. Methods: A retrospective IRB-approved study was conducted on patients who underwent renal mass biopsy for histopathologic diagnosis at a single center from 2015 to 2021. Patients with a prior history of malignancy or a renal mass >4 cm were excluded. Descriptive statistics were used to summarize patient demographics, tumor size, the imaging modality used for biopsy, procedure details, complications, pathological diagnosis, and post-biopsy management. A biopsy was considered successful when the specimen was sufficient for diagnosis without need for a repeat biopsy. Complications were graded according to the SIR classification of adverse events. A chi-squared test (significance level set at *p* ≤ 0.05) was used to compare the success rate of biopsies in different lesion size groups. Results: A total of 167 patients met the inclusion criteria. The median age was 65 years (range: 26–87) and 51% were male. The median renal mass size was 2.6 cm (range: one–four). Ultrasound was solely employed in 60% of procedures, CT in 33%, a combination of US/CT in 6%, and MRI in one case. With on-site cytopathology, the median number of specimens obtained per procedure was four (range: one–nine). The overall complication rate was 5%. Grade A complications were seen in 4% (*n* = 7), consisting of perinephric hematoma (*n* = 6) and retroperitoneal hematoma (*n* = 1). There was one grade B complication (0.5%; pain) and one grade D complication (0.5%; pyelonephritis). There was no patient mortality within 30 days post-biopsy. Biopsy was successful in 88% of cases. A sub-group analysis showed a success rate of 85% in tumors <3 cm and 93% in tumors ≥3 cm (*p* = 0.01). Pathological diagnoses included renal cell carcinoma (65%), oncocytoma (18%), clear cell papillary renal cell tumors (9%), angiomyolipoma (4%), xanthogranulomatous pyelonephritis (1%), lymphoma (1%), high-grade papillary urothelial carcinoma (1%), and metanephric adenoma (1%), revealing benign diagnosis in 30% of cases. The most common treatment was surgery (40%), followed by percutaneous cryoablation (22%). In total, 37% of patients were managed conservatively, and one patient received chemotherapy. Conclusion: This study demonstrates the safety and diagnostic efficacy of image-guided biopsy of small renal masses. The diagnostic yield was significantly higher for masses 3–4 cm in size compared to those <3 cm. The biopsy results showed a high percentage of benign diagnoses and informed treatment decisions in most patients.

## 1. Introduction

Renal masses pose a considerable clinical challenge due to their diverse differential diagnosis, varying prevalence, and potential impact on patient mortality. Renal cell carcinomas (RCCs) represent the predominant subtype of primary renal neoplasms, accounting for 80–85% of cases, and their diagnosis is associated with a 35% mortality rate within 5 years of diagnosis [1] and less than 10% for metastatic RCC [2]. In contrast, benign renal tumors constitute approximately 15–20% of renal neoplasms, encompassing entities such as oncocytoma and angiomyolipoma [3,4].

In recent decades, the incidence of kidney cancer diagnoses has increased significantly, which is primarily attributed to the widespread adoption of axial imaging techniques. This technological advancement has enabled the earlier detection of cancerous lesions even before the onset of noticeable symptoms [5]. The most substantial increase in incidence has been observed in small, localized tumors. These tumors, characterized by their confined presence within the kidney without any evidence of local spread, lymph node involvement, or distant metastases, now constitute a noteworthy proportion, accounting for more than 40% of all kidney cancers [6,7].

The size of a renal mass (RM) is directly proportional to the probability of the mass being malignant, with a 50% malignancy risk in RMs ≤ 1 cm and up to 75% in lesions 1 to 2.9 cm in size [8]. Small renal masses (SRMs) (≤4 cm) have often been treated with surgery (partial and radical nephrectomies) or cryoablation without the use of biopsy before the treatment [9,10]. Several studies have compared various cryoablation techniques and contrasted them with surgical treatments for small renal masses [11,12]. While image-guided renal mass biopsy (RMB) is gaining increased diagnostic acceptance, there are limited data concerning the safety and diagnostic yield of biopsy for SRMs.

This study aims to address this knowledge gap by analyzing the safety and efficacy of RMBs, specifically in cases of SRMs. The hypothesis posits that RMB is not only a safe procedure but also highly diagnostic, optimizing treatment planning in cases of SRMs measuring 4 cm or smaller.

## 2. Materials and Methods

### 2.1. Study Population

A retrospective, single-center study was conducted to evaluate the safety and efficacy of image-guided biopsy for SRMs between January 2015 and December 2021 in the United States. The study was Health Insurance Portability and Accountability Act-compliant and approved by the institutional review board (IRB) at Northwestern University. The IRB waived the requirement for individual patient informed consent. This waiver is in accordance with ethical guidelines for retrospective studies, ensuring that patient confidentiality and data integrity were maintained throughout our research. Ethical considerations were adhered to throughout the study, with all patient data anonymized, stored in password-encrypted files, and treated in accordance with relevant privacy regulations. It included patients with imaging-diagnosed RMs ≤4 cm who underwent RMBs by fellowship-trained interventional radiologists. Patients with renal masses >4 cm or a history of other malignancies to avoid confusion with metastasis were excluded from the study. A total of 167 patients were included in the study, while 61 were excluded due to not meeting defined criteria: 42 of these excluded patients had a history of other malignancies, and 19 had RMs larger than 4 cm. Eligible patients were identified by meticulously reviewing available medical records from the specified study period based on the fulfillment of our specific inclusion and exclusion criteria on retrospective reviews of consecutive patients. Our inclusion criteria were determined by the initial imaging diagnosis, as noted in medical records, and the tumor size, as defined in baseline imaging. To ensure a robust and unbiased patient selection process, stringent measures were taken in the identification of eligible candidates. This involved a thorough examination of medical records from the specified timeframe, where consecutive patients were evaluated against predetermined criteria, minimizing the potential for inherent biases in the selection process.

### 2.2. Biopsies

The prebiopsy evaluation included history, a physical examination, and laboratory (platelet count and coagulation studies) and imaging studies. Platelet counts >50,000/mm and INR < 1.5 were insured at the time of biopsy. Biopsies were performed under ultrasound (US), computed tomography (CT) scans, a combination of US and CT scan, or magnetic resonance imaging (MRI) guidance. The operator-selected needle type and size used and the number of tissue samples obtained were recorded. Rapid on-site evaluation (ROSE) of touch preparations of the biopsy tissue were performed based on cytopathology in each case during the biopsy procedure. After the procedure, the patients were monitored in the recovery room, and any immediate complications were recorded. The patients were discharged home on the same day as the procedure. The success of a biopsy was defined as obtaining sufficient tissue for a diagnosis, as confirmed by an on-site pathologist performing a ROSE, without the need for a repeat procedure. A biopsy was considered unsuccessful if it revealed normal renal parenchyma or insufficient tissue for diagnosis.

### 2.3. Post-Biopsy Renal Mass Management

The post-biopsy management following the completion of the biopsy and the subsequent availability of the pathological diagnosis was recorded. The recorded management options included surgery, percutaneous cryoablation, chemotherapy, or conservative management.

### 2.4. Analysis

The patients’ demographics, the tumor size and location, the imaging modality used, procedure details, complications, pathological diagnosis, and subsequent treatments were collected and analyzed using descriptive statistics. To further investigate potential differences in safety and efficacy based on lesion size as a surrogate for technical difficulty, the patients were stratified by RM size into two groups: masses ≥3 cm and masses <3 cm. Complications resulting from the biopsy were graded according to the Society of Interventional Radiology (SIR) classification of adverse events [13]. Statistical analyses were performed using the SPSS statistical software version 27 and *p*-values less than 0.05 were considered statistically significant.

## 3. Results

### 3.1. Study Population and Mass Characteristics

A cohort of 167 patients fulfilled the predefined inclusion criteria and were included in this study. The median age of the participants was 65 years, with a range of 26 to 86 years. Among the enrolled patients, 86 (51%) were male, while 81 (49%) were female. Out of the 71 patients with available Eastern Cooperative Oncology Group (ECOG) performance status, 51 had a grade 0 status, 12 had a grade 1 status, 4 had a grade 2 status, 3 had a grade 3 status, and 1 had a grade 4 status. The median size of the RMs was 2.6 cm, varying from 1 to 4 cm. The distribution of masses revealed that 43% were located in the right kidney, whereas 57% were found in the left kidney. The majority of RMs (38%) were situated at the lower pole of the kidney, followed by 33% in the interpolar region and 29% at the upper pole.

### 3.2. Biopsies

The demographic data and mass and biopsy characteristics are summarized in Table 1. Concerning the imaging technique used during the procedure, US was solely employed in 101 cases (60%), CT scan in 55 cases (33%), a combination of US/CT in 10 cases (6%), and MRI in 1 case (1%). In most cases (87%), patients were positioned prone, while a smaller proportion (13%) were positioned supine. In terms of needle selection, a Temno needle (Merit Medical System, South Jordan, UT, USA) was utilized in 61% of the procedures, a BioPince (Argon Medical Devices, Plano, TX, USA) needle in 18% of the cases, and both Temno and BioPince needles were used in 7% of the procedures. The median needle size employed was 18 Gauge (G), ranging between 16 and 20 G. The median number of specimens obtained per procedure was four, ranging between one and nine specimens per procedure.

### 3.3. Safety

Complications occurred in nine biopsy procedures, resulting in an overall complication rate of 5%. Among these cases, grade A complications were observed in seven instances (4%), consisting of six cases of perinephric hematoma and one case of retroperitoneal hematoma. Additionally, there was one grade B complication (0.5%) characterized by pain and one grade D complication (0.5%) involving pyelonephritis. The post-biopsy complications are summarized in Table 2. There was no occurrence of mortality within 30 days post-biopsy. Table 3 summarizes characteristics of cases in which complications were observed. Upon stratification based on the size of the masses, complications were observed in 5 out of 60 patients (8%) in the group with mass sizes ≥3 cm. Within this sub-group, three cases displayed grade A complications (5%) in the form of perinephric hematoma, along with one case each of a grade B (1.5%) complication (pain) and a grade D (1.5%) complication (pyelonephritis). In contrast, complications were observed in 4 out of 107 patients (3%) with masses <3 cm. These cases included four instances of grade A complications (2%), characterized by perinephric hematoma, along with an additional case (1%) of retroperitoneal hematoma. A chi-squared test demonstrated no significant difference in the complication rate (8% in masses ≥3 cm vs. 3% in masses <3 cm) between these two size groups (*p* = 0.1).

### 3.4. Technical Success

Out of a total of 167 patients, a biopsy was successfully conducted in 147 cases (88%). The remaining 20 patients (12%) had unsuccessful biopsies without confirmed tissue diagnosis. The biopsies of seven cases initially indicating normal renal parenchyma were subsequently repeated, revealing the presence of RCC in all instances. Out of the remaining cases with unsuccessful biopsies, nine were managed conservatively, two underwent nephrectomy, and two were lost to follow-up. After stratification, the results revealed a success rate of 93% (56 out of 60 patients) for tumors measuring ≥3 cm, whereas tumors measuring <3 cm exhibited a success rate of 85% (91 out of 107 patients). A chi-squared test was performed, which demonstrated a significant difference in the success rate between these two size groups (*p* = 0.01). An extensive sub-group analysis incorporating tumor side and location, patient position, needle size, and imaging modality employed during the procedure revealed that there was no significant difference in the success rate among these specific sub-groups (Table 4).

The pathologic diagnoses from the biopsy samples encompassed a diverse range of findings. RCC was the most prevalent, accounting for 95 cases (65%). Oncocytoma was identified in 27 cases (18%), while a clear cell papillary renal cell tumor was observed in 14 cases (9%), angiomyolipoma was observed in 6 cases (4%), and lymphoma in 2 cases (2%). One case each of xanthogranulomatous pyelonephritis (1%), high-grade papillary urothelial carcinoma (1%), and metanephric adenoma (1%) were detected. Post-biopsy diagnoses are summarized in Table 5. Note that, according to the WHO 2022 Classification of Kidney Tumors, the name “clear cell papillary renal cell carcinoma” was changed to “clear cell papillary renal cell tumor” [14]. Consequently, clear cell papillary renal cell tumor is not classified as a subtype of RCC.

### 3.5. Post-Biopsy Renal Mass Management

The most common treatment employed after biopsy was surgery, accounting for 40% (60 cases) of the cases. Among the surgical interventions, partial nephrectomies were performed in 63% of cases. Radical nephrectomies constituted the remaining 37% of surgical treatments. In 95% of cases (57/60 cases treated with surgery), the post-surgery diagnosis was consistent with the biopsy diagnosis. Concerning the discordant results, all three cases were diagnosed as papillary-type RCC on biopsy. A post-surgery pathological analysis showed two cases of clear cell papillary renal cell tumors and one case of clear cell RCC. Percutaneous cryoablation was the second most common treatment approach, utilized in 22% of patients (34 cases). In total, 37% of individuals (55 cases) opted for conservative management, while a single (1%) patient underwent chemotherapy; 17 patients were lost to follow-up. In our cohort of conservatively treated RCC patients, the histological analysis revealed that 53% were diagnosed as clear cell renal cell carcinoma, followed by 41% classified as papillary RCC and 6% as chromophobe RCC. Four cases of clear cell papillary renal cell tumors were treated conservatively.

## 4. Discussion

The increased detection of SRMs through imaging combined with limited data on the safety and diagnostic yield of biopsy for small renal masses presents a clinical challenge for subsequent management. A retrospective review of 167 patients undergoing biopsy for SRM over a 6-year period revealed a complication rate of 5%, mostly grade A, managed conservatively, and an overall success rate of 88%. Following the sub-group analysis based on mass size, patients with renal masses <3 cm had lower success rates but similar complication rates. In post-biopsy management, 40% of patients underwent surgery, while a considerable 37% received conservative management.

This experience revealed the safety of SRM biopsy, with a total of nine complications observed. Among these, only one major complication (grade D) occurred, which involved pyelonephritis requiring >48 h of hospitalization and antibiotic therapy. The remaining complications were minor (grade A and B), including self-limited hematomas and pain that required minimal or no treatment and had no lasting effects. This low complication rate aligns with existing published research on renal mass biopsies. Alle et al. reported a complication rate of 7.6% in a cohort of 169 patients, with one major complication involving retroperitoneal hematoma requiring transfusion and arterial embolization of a post-procedure arteriovenous fistula [15]. Similarly, Richard et al. found a low rate of adverse events (8.5%) in a retrospective study of 529 patients, with most complications being self-limited except for one perirenal hematoma that required embolization [16]. A retrospective study including 108 patients who underwent 183 office-based US-guided percutaneous RMBs with a median mass size of 3.3 cm showed only three grade I Clavien–Dindo surgical complications, all of which were managed conservatively [17,18]. These findings, along with the literature, reinforce the low rate of severe complications associated with image-guided renal biopsies, such as significant blood loss necessitating transfusion (0.4%) or kidney loss. These data provide reassurance regarding the safety profile of the procedure and support its use as a valuable diagnostic tool for SRMs [19].

The diagnostic success rate of 88% is consistent with the upper range of the diagnostic rates reported in the sparse literature, which range from 70% to 91% for biopsies conducted on SRMs [16]. In a systemic review conducted by Patel et al., encompassing 20 studies with 2979 patients and over 3000 RMBs, an overall nondiagnostic rate of 14.1% was observed [20]. In a similar fashion, Marconi et al. reviewed 57 studies involving 5228 patients, revealing a diagnostic rate of 92% for RMBs, but this included masses >4 cm [21].

Tumor seeding along the biopsy tract is another concern that may contribute to hesitance to pursue RMB. However, the available literature reports only a limited number of cases, with a total of 16 reported cases found in a recent review [22,23]. In a large meta-analysis including 57 studies and 5228 patients, only one case of tumor seeding was reported [21]. Factors that may be associated with tumor seeding include certain tumor subtypes, multiple punctures of the RM, ≥20-gauge needles, and the lack of a coaxial technique [24]. It is important to note that similar occurrences of tumor seeding have also been documented in several other types of biopsies, for example, liver and lung biopsies. A comprehensive review of the literature reveals that the observed incidence of tumor seeding after biopsy of hepatocellular carcinoma ranges from 0.6% to 5.1% of cases [25]. Furthermore, there have been reports of lung cancer tumor seeding with chest wall implantation in several documented cases [26,27]. Therefore, tract seeding is not unique to renal mass biopsy, and prior experience is largely anecdotal, with few reported cases.

Solid RMs have historically been assumed to be RCC and surgical resection was performed without prior tissue diagnosis. However, it has been recognized that a significant proportion of nephrectomies are performed for benign conditions [15]. Kim et al., in a cohort including 18,060 patients between 2007 and 2014, showed an overall prevalence of benign pathologic findings after partial nephrectomy of 30.9% [28]. Additionally, approximately 20–25% of SRMs, which are defined as imaging-detected tumors ≤4 cm in diameter, are benign renal tumors, such as oncocytomas, angiomyolipomas, and metanephric adenomas. Yet, biopsy of renal masses has been underutilized. In a study by McClure et al., of 20,107 patients with RCC, only 1012 (7.3%) had a biopsy performed [10]. In another study by Maurice et al., of 171,406 RCC patients eligible, 21,019 (12.3%) were biopsied [9]. This study demonstrated that 24% of cases in our cohort yielded a benign diagnosis upon pathological examination, with oncocytoma being the predominant finding. Additionally, among the patients managed conservatively, comprising 37% of the total cohort, 47% were diagnosed as benign.

The American Urological Association (AUA) guidelines have developed an algorithm to guide the clinical application of RMBs. These guidelines acknowledge the considerable variation in practice patterns. Due to a scarcity of sufficient evidence in the current literature, most of the guideline statements concerning RMBs rely on clinical principles and expert opinions rather than explicit recommendations [29,30]. Generally, RMBs may not be necessary for patients who have already opted for conservative management or surgical intervention, irrespective of the biopsy results. However, there are specific scenarios where RMBs should be considered to potentially impact clinical decision making. This includes patients with severe chronic kidney disease, as their management can be more complex. Additionally, RMBs are advised for cases where the choice between partial and radical nephrectomy is challenging, and further risk stratification could be beneficial [20,31,32]. In parallel with other published studies, the present data demonstrate that this minimally invasive procedure has a significant impact on post-biopsy management decisions, specifically in reducing nephrectomies for benign etiologies that were previously performed without prior biopsy. Consequently, there exists the potential to improve patient outcomes, reduce healthcare costs, and spare individuals from the potential physical and psychological burdens associated with unnecessary surgeries. With these motivations, there is a compelling need for the integration of updated guidelines that advocate for a broader utilization of RMBs. The evidence supports the notion that incorporating RMBs into diagnostic and management algorithms can yield valuable information for risk stratification and treatment decision making in patients with SRMs. Therefore, it is crucial for new guidelines to reflect the growing body of evidence and recognize the importance of RMBs as a valuable tool in clinical practice, especially when ROSE by cytopathology is possible. ROSE during renal mass biopsy enables the immediate assessment of specimen adequacy and appropriate specimen triage. Furthermore, ROSE can ensure sufficient tissue for immunohistochemistry or molecular testing for a better characterization of RMs. The real-time evaluation helps in determining the need for additional sampling or interventions, thus reducing the requirement for repeat procedures. The available literature shows the high sensitivity and specificity of on-site cytopathology, leading to timely treatment planning and improved patient outcomes [22,33,34].

This study’s strengths include a substantial number of contemporary cases. ROSE based on cytopathology is a major strength of our workflow, allowing for an immediate review of specimen adequacy. However, there are limitations to this study, including the single-center retrospective nature of the data collection and the lack of a control group that limit the ability to draw definitive conclusions about the efficacy and safety of SRM biopsy, as comparisons to standard practices or alternate methods are not available. However, given the current scarcity of the literature in this area, this study provides important evidence for future research studies on expected outcomes after renal biopsy. The insights gained here can serve as a preliminary guide for larger prospective studies and institutional safety benchmarks. Such studies would not only mitigate the biases present in retrospective analyses but also provide a clearer comparative framework against standard or alternative diagnostic procedures. Additionally, this study did not explicitly detail the clinical scenarios or indications for renal mass biopsy, potentially affecting the interpretation of the findings. The current study does not include renal masses <1 cm and offers no information regarding biopsies for smaller masses.

## 5. Conclusions

These data from a retrospective study demonstrate the safety and diagnostic efficacy of SRM biopsies, with a high success rate and acceptable complication profile. The results provide useful information, enriching current practice guidelines and supporting clinical decision making in the evaluation and management of patients with SRMs.

## Figures and Tables

**Table 1 cancers-16-00835-t001:** Patient demographics and renal mass and biopsy characteristics.

Features	Number (%)
Number of patients included	167
Median age (range) (years)	65 (26–86)
Gender	
Male	86 (51%)
Female	81 (49%)
Median size (range) (cm)	2.6 (1–4)
Mass side	
Left kidney	95 (57%)
Right kidney	72 (43%)
Mass location	
Lower pole	63 (38%)
Interpolar	55 (33%)
Upper pole	49 (29%)
Patient position	
Prone	145 (87%)
Supine	22 (13%)
Imaging technique used	
US	101 (60%)
CT	55 (33%)
US/CT	10 (6%)
MRI	1 (1%)
Needle type	
Temno	102 (61%)
BioPince	31 (18%)
Temno/BioPince	11 (7%)
Not documented	23 (14%)
Median number of specimens (range)	4 (1–9)

**Table 2 cancers-16-00835-t002:** Post-biopsy complications.

Complications	Number (%)
Perinephric hematoma (Grade A)	6 (3%)
Retroperitoneal hematoma (Grade A)	1 (1%)
Pain (Grade B)	1 (1%)
Pyelonephritis (Grade D)	1 (1%)

**Table 3 cancers-16-00835-t003:** Characteristics of cases in which complications were observed.

Features	Number (%)
Number of patients	9
Median size (range) (cm)	4 (1.6–4)
Mass side	
Left kidney	5 (55%)
Right kidney	4 (45%)
Mass location	
Lower pole	4 (44%)
Interpolar	3 (33%)
Upper pole	2 (23%)
Patient position	
Prone	7 (78%)
Supine	2 (22%)
Imaging technique used	
US	4 (44%)
CT	3 (33%)
US/CT	2 (23%)
Needle type	
Temno	3 (33%)
BioPince	2 (23%)
Temno/BioPince	3 (33%)
Not documented	1 (11%)
Median number of specimens (range)	2.5 (1–5)
Anticoagulation use	
Heparin	1 (11%)
Apixaban	1 (11%)
None	7 (78%)

**Table 4 cancers-16-00835-t004:** Sub-group analysis of renal mass and biopsy characteristics.

Stratification Criteria	Success	Chi-Square Test *p*-Value
Mass side (left/right)	84 (88%)/67 (93%)	0.3
Mass location (upper/interpolar/lower)	47 (96%)/48 (87%)/56 (89%)	0.2
Patient position (prone/supine)	133 (95%)/18 (90%)	0.2
Imaging modality (US/CT/US/CT)	92 (93%)/49 (89%)/9 (82%)	0.5
Needle size (18 G/20 G/18/20 G)	76 (88%)/53 (95%)/14 (93%)	0.5

**Table 5 cancers-16-00835-t005:** Pathologic diagnosis of the small renal masses.

Pathologic Diagnosis	Number (%)
RCC	95 (65%)
Clear cell (Low Fuhrman’s nuclear grade 74% (32/43 available grades); High Fuhrman’s nuclear grade 26% (11/43 available grades)	52 (55%)
Papillary (Low Fuhrman’s nuclear grade 65% (15/23 available grades; High Fuhrman’s nuclear grade 35% 8/23 available grade)	35 (37%)
Chromophobe	8 (8%)
Oncocytoma	27 (18%)
Clear cell papillary renal cell tumor	14 (9%)
Angiomyolipoma	6 (4%)
Xanthogranulomatous pyelonephritis	1 (1%)
Lymphoma	2 (1%)
High-grade papillary urothelial carcinoma	1 (1%)
Metanephric adenoma	1 (1%)

## Data Availability

The data presented in this study are available on request from the corresponding author. The data are not publicly available due to privacy and ethical considerations.

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
