# Peer review of "Safety and Diagnostic Efficacy of Image-Guided Biopsy of Small Renal Masses"

_cancers, 2024, doi:10.3390/cancers16040835_

Round 1
Reviewer 1 Report
Comments and Suggestions for Authors
I have reviewed the manucript "Safety and Diagnostic Efficacy of Image-Guided Biopsy of Small Renal Masses", which aims to evaluate the safety, diagnostic yield, and management after image-guided percutaneous biopsy for small renal masses. The topic is relevant. The manuscript is well-written, organized, easy to follow.
Major concerns:
- retrospective study. I am not sure how this study adds to currently available evidence.
- retrospective study. It is not clear to me how were the patients looked up to be considered for elegibility. I am concerned about potential bias. Did you start by looking up the biopsies performed in the study period?
- the original indications (or clinical scenarios) for renal biopsy of these patients are actually not clear, which limits interpretation of the results reported. Even if they authors showed that renal biopsy is safe, one would still not be interested in performing it in patients with a clear RCC by CT, because it would not change management. As the Introduction explains and Discusion further comments, the relevant clinical issue concerning renal biopsies is whether they could thereof change management (e.g., avoid total or partial nephrectomy, etc). Yet, the manuscript did not really analyze this issue. One would like to know which were the presumptive imaging diagnoses of the renal masses / which were the clinical indications for biopsy (i.e., which were, for these patients, the (line 243): "specific scenarios where RMB should be considered to potentially impact clinical decision-making"). All the analyses of safety, diagnostic yield and management should be shown by subgroups of such scenarios to be truly informative to the readership in clinical practice.
Other important concerns:
- according to the introduction, there is "50% malignancy risk in RM ≤ 1cm". The minimum size of RM studied by the authors over a span of 6 years is 1.6 cms. I wonder what is the incidence of <1.6 cm renal lesions. I am concerned about potential bias. It seems unlikely that over 6 year there was not a single RM <1.6 cm.
- did the participants signed a written informed consent? or did the IRB waived the need of an IC?
- applicability of the findings: ROSE is not widely available, so I wonder whether the results reported in this manuscript would change in conditions more similar to widely daily practice.
- by "clear cell papillary renal cell tumor" do you mean "clear cell papillary renal cell carcinoma"? if that is the case, please note that it should be included within the RCC group according to the WHO Classification of Tumours of the Urinary System and Male Genital Organs.
- I did not understand the following sentence: "Study strengths include a substantial number of contemporary cases over an extended follow-up period post-biopsy", or how this could be considered a strength for the study reported in this manuscript.
Reviewer 2 Report
Comments and Suggestions for Authors
In my opinion, a well-written paper that deserves to be published. The complications of RMB described (5%) are minor overall. The authors are to be congratulated for this. Nevertheless, these must be outweighed by the benefit of the resulting optimized treatment strategy. As indicated in the discussion, there may be clinical scenarios in which the histological diagnosis influences the further course of action. However, this is unlikely to apply to most cases.
The authors correctly state that the majority of SRMs are renal cell carcinomas (68%), followed by oncocytomas (17%). The treatment of T1a tumours (<4 cm) - malignant or benign - is not really different and today consists of a tumour resection that preserves the kidney as far as possible. Surgical resectability generally depends on the size and location or organ-invasive growth of the tumour and can now be assessed in advance using modern high-resolution imaging techniques. What influence does knowledge of the histology have on the treatment? The work presented here could therefore benefit considerably if the authors were to draw up an algorithm at the end of the discussion for which clinical constellations it really makes sense to carry out an RMB that influences the therapeutic procedure.
Reviewer 3 Report
Comments and Suggestions for Authors
1. The article provides valuable insights into the safety and efficacy of image-guided biopsy for small renal masses. To enhance its potential for publication, consider incorporating recent literature, providing more context, and addressing specific suggestions mentioned above.
2. Introduction:
- The introduction is well-structured, providing a clear background on the clinical challenge of renal masses. However, the mortality rate statistic could be referenced more specifically.
- The literature review lacks recent references, and it would benefit from including the latest studies to establish the novelty and relevance of the research. For example, your research could benefit from including the citation (PMID: 36286402) to further delve into the comparison between ablative techniques in this multi-centric analysis and to contextualize your analysis within the current landscape of research on the treatment of small renal masses. Include also the following (PMID: 36216659) to compare ablative techniques vs surgery
3. Materials and Methods:
- The study period is clearly defined. However, more information on the geographical location of the single-center study would enhance context.
- Ethical considerations are briefly mentioned; expanding on the ethical approval process and measures taken for patient confidentiality would strengthen this section.
- Exclusion criteria are well-defined, but it would be helpful to explain the rationale behind excluding patients with a history of other malignancies.
- The biopsy techniques are adequately described, but details on the qualifications and experience of the operators could be included for transparency.
- The success criteria for biopsy are well-defined. However, the definition of "sufficient tissue for diagnosis" could be clarified, as it might vary among pathologists.
4. Results:
- The demographics and characteristics of the study population are well-presented. However, it would be beneficial to provide more context on how this cohort compares to existing literature.
- Complication rates and their categorization are provided, but a discussion on how these rates compare with other studies would strengthen the interpretation.
- The discussion on stratification by mass size is informative, but statistical significance alone may not fully convey clinical relevance. Consider discussing the clinical implications of these findings.
- Pathological diagnoses are well-summarized, but a brief discussion on the clinical significance of these diagnoses could be included.
- The post-biopsy management section is concise but could benefit from a more detailed discussion on the rationale behind the chosen treatments.
5. Discussion:
- The discussion provides a comprehensive overview but could be more focused on directly addressing the study's objectives and how the results contribute to existing knowledge.
- A more critical analysis of the limitations, such as the retrospective nature of the study and lack of a control group, would strengthen the discussion.
- The discussion could highlight the practical implications of the study's findings and suggest avenues for future research.
6. Conclusion:
- The conclusion is concise but could be expanded to include a summary of key findings and their potential impact on clinical practice.
Round 2
Reviewer 1 Report
Comments and Suggestions for Authors
The authors have made an effort to reply to my comments. As a clinical researcher I am aware of the many difficulties that one may encounter when performing a clinical study like the one they report in the submitted manuscript. As a reviewer it is not my primary aim to ask for explanations but to help accommodating the piece of information provided in the form of a manuscript to be published, which concerns the underlying study, so that future readership with similar observations, questions and comments may have them already solved in the manuscript.
To my best consideration, the authors' reply to my comments provide explanations, yet I had difficulties seeing that translated into revisions of the manuscript that attend the comments and accommodate the manuscript accordingly, thus clarifying theses issues to the readership in the manuscript:
- what this study adds to currently available evidence
- how were the patients looked up to be considered for eligibility, and whether this could lead to potential bias
- the original indications --> there should be any available. It would be a rare practice to perform an interventional procedure withouth any clinical indication (at least pressumptive diagnosis).
- what is the potential bias introduced by not including renal masses <1.6 cms
- whether the IRB waived the IC
- informing a relevant recent change in the WHO classification, pointed out by the authors to me
In a potential following round, revision of the revised manuscript accommodating theses issues would also be facilitated by the authors with indication of the page and line where they have performed the corresponding editions.
Reviewer 3 Report
Comments and Suggestions for Authors
The manuscript has been significantly improved compared to previous versions and is now an engaging and informative read. The topics discussed are extremely interesting, and the author has presented a unique and well-documented perspective.
Author Response
Thank you for acknowledging the improvements. Your previous feedback played a crucial role in shaping this manuscript, and I'm pleased that the engaging perspective on these interesting topics has been well-received.
Round 3
Reviewer 1 Report
Comments and Suggestions for Authors
The authors now pointed out in the rebuttal where in the manuscript can be found the editions they comment on in consideration of my observations.
***Not having the clinical indication of the procedures is a major limitation of this study that should be acknowledge in the abstract if this study is to be published because then the readership will be rapidly inform. Thus, they can decide whether it is worth going over the data reported in this manuscript.
***Once more I emphasize that not having a clear clinical indication for performing the procedures seems a rare practice to me, which makes me wonder whether (as a potential reader) I would be interested in proceeding further with going over the manuscript.
***in relation to the previous comment, I also think that concerning interventional procedures, it may be even ethically questionable that there is no clear clinical indication for performing the procedures.
Please note that most pages and lines indicated in the rebuttal seem to be incorrect (which makes my work as a reviewer more difficult); in particular I could not find the change made in relation to my comment concerning potential bias introduced by not including renal masses <1.6 cms. ("page 3, line 143").
Also, the code of the waiver concerning the IRB approval is still not provided in the manuscript. Particularly taking into account my main comments of this review report, I think this code should be incorporated if the manuscript is to be published (I see no reason for not providing it still in the 3rd round of revision).
Comments on the Quality of English Language
I am concerned that not having a clinical indication for performing the procedures is a major limitation of this study. I am doubtful how this study may add to literature with this critical information to interpret the results. This issue may also turn this study questionable because interventional procedures were performed in patients without clear clinical indication.
Two rounds of revisions providing sloppy rebuttal letters, in addition to not having such vital data (mentioned above) make me wonder of the minimum acceptable rigurosity of the authors for performing a clinical study.
